# AFF4 binding to Tat-P-TEFb indirectly stimulates TAR recognition of super elongation complexes at the HIV promoter

Ursula Schulze-Gahmen[1]*, Huasong Lu[1], Qiang Zhou[1], Tom Alber[1,2]

[1]Department of Molecular and Cell Biology, University of California, Berkeley, Berkeley, United States; [2]California Institute for Quantitative Biosciences, QB3, University of California, Berkeley, Berkeley, United States

**Abstract** Superelongation complexes (SECs) are essential for transcription elongation of many human genes, including the integrated HIV-1 genome. At the HIV-1 promoter, the viral Tat protein binds simultaneously to the nascent TAR RNA and the CycT1 subunit of the P-TEFb kinase in a SEC. To understand the preferential recruitment of SECs by Tat and TAR, we determined the crystal structure of a quaternary complex containing Tat, P-TEFb, and the SEC scaffold, AFF4. Tat and AFF4 fold on the surface of CycT1 and interact directly. Interface mutations in the AFF4 homolog AFF1 reduced Tat–AFF1 affinity in vivo and Tat-dependent transcription from the HIV promoter. AFF4 binding in the presence of Tat partially orders the CycT1 Tat–TAR recognition motif and increases the affinity of Tat-P-TEFb for TAR 30-fold. These studies indicate that AFF4 acts as a two-step filter to increase the selectivity of Tat and TAR for SECs over P-TEFb alone.

*For correspondence: uschulze-gahmen@berkeley.edu

**Competing interests:** The authors declare that no competing interests exist.

## Introduction

Transcription of the HIV genome by RNA polymerase II (Pol II), like the expression of many cellular genes, is largely regulated at the step of transcript elongation. (*Levine, 2011*; *Lin et al., 2011*; *Luo et al., 2012*; *Zhou et al., 2012*). Pol II is recruited to the HIV promoter and initiates transcription, which stalls after a 30–50 nucleotide transcript containing the trans-activating response region (TAR) is formed. The HIV Tat protein bound to a host super elongation complex (SEC) recognizes TAR and releases the paused polymerase (*He et al., 2010*; *Sobhian et al., 2010*). It is yet unclear how TAR and Tat specifically recruit SECs in preference to other complexes in the cell that contain SEC subunits.

HIV Tat binds simultaneously to TAR and positive elongation factor b (P-TEFb), composed of CDK9 and Cyclin T1 (CycT1) subunits. In turn, P-TEFb and the transcriptional elongation factors ELL2 and ENL/AF9 associate with hydrophobic segments in the approximately 1200-amino-acid AFF1 or AFF4 scaffold, together forming the SEC (*He et al., 2010*; *Lin et al., 2010*; *Sobhian et al., 2010*). P-TEFb triggers promoter escape by phosphorylating two negative elongation factors (DSIF and NELF), as well as the C-terminal domain (CTD) of Pol II (*Ott et al., 2011*; *Zhou et al., 2012*; *Lu et al., 2013*). In contrast, ELL2 is thought to stimulate Pol II processivity (*Shilatifard et al., 1997*) and ENL/AF9 appears to bridge the SEC to RNA polymerase II-associated factor complexes (*He et al., 2011*). Overexpression of an AFF1 fragment that binds to P-TEFb, but not the other components of SECs, has a strong inhibitory effect on HIV transcription, indicating that productive HIV transcription requires a complete SEC (*Lu et al., 2013a*).

As a key regulator of transcription, P-TEFb itself is regulated through complex formation with other proteins and RNA. Recent studies have shown that the CycT1 subunit tightly associates with AFF1 or

**eLife digest** The rate at which many genes are expressed as proteins depends on a process called transcriptional elongation. This process takes place as the region of DNA that defines the gene is transcribed into an RNA molecule, and it is catalyzed by an enzyme called RNA polymerase II. However, this process often stalls shortly after it starts, and another enzyme called a positive transcription elongation factor is needed to restart it.

The human immunodeficiency virus (HIV) is a retrovirus that hijacks the gene expression machinery inside immune cells in order to replicate itself. To do this as efficiently as possible, the elongation factor needs to restart the transcription process as quickly as possible. To ensure that this happens the virus produces a protein called Tat that binds to the short region of RNA that has already been made. At the same time the Tat protein also combines with other proteins to form a multi-protein machine called the super elongation complex. Other proteins in the super elongation complex include a 'scaffold' protein called AFF4, a positive elongation factor called P-TEFb, and at least two additional transcription factors.

Until recently researchers did not know how the Tat protein was able to recruit super elongation complexes to the correct location without recruiting other complexes that contained similar protein subunits. Now Schulze-Gahmen et al. have shed new light on this mystery by working out the crystal structure of the complex formed by the elongation factor P-TEFb when it forms a complex with the Tat protein and a scaffold protein called AFF4.

The results show that direct interactions between the Tat and scaffold proteins help to recruit the super elongation complex to the correct location. The three-way interactions between Tat, AFF4, and P-TEFb form a binding surface that encourages the complex to bind to the RNA. Overall, Schulze-Gahmen et al. show that the super elongation complex is much more likely to be recognized by the Tat protein and then bind to RNA than just the elongation factor on its own.

---

AFF4 and that the ternary complex of CDK9, CycT1, and AFF1 (*He et al., 2010*; *Lin et al., 2010*) moves between various active and inactive P-TEFb complexes (*Lu et al., 2014*). These assemblies include the SECs (*He et al., 2010*; *Lin et al., 2010*), a complex with Brd4 (*Jang et al., 2005*; *Yang et al., 2005*), and the inhibitory 7SK snRNP (*Lu et al., 2013a*; *Yang et al., 2001*; *Yik et al., 2003*). The existence of various P-TEFb complexes in the cell raises the question of how Tat and TAR discriminate among these functionally diverse assemblies.

An initial clue to the origin of the specificity of Tat and TAR for recognizing SECs was provided by the crystal structure of the AFF4-P-TEFb complex (*Schulze-Gahmen et al., 2013*). In this complex, the intrinsically disordered AFF4 fragment was folded on the surface of CycT1 in an orientation adjacent to the Tat binding site. A model of the quaternary Tat-AFF4-P-TEFb complex, based on the superposition of AFF4-P-TEFb and Tat-P-TEFb (*Tahirov et al., 2010*), predicted direct interactions between Tat and AFF4 (*Schulze-Gahmen et al., 2013*). These direct contacts were proposed to account for an 11-fold increase in the affinity of Tat for P-TEFb in the presence of AFF4.

To better understand the structural basis for the critical role of AFF1/4 in HIV transcription, we determined the crystal structure of P-TEFb in complex with Tat and AFF4. The Tat-AFF4-P-TEFb structure and in vivo reporter assays in HeLa cells confirm the direct Tat–AFF4 interactions. In addition, AFF4 contacts CycT1 residues that are part of the flexible Tat–TAR recognition motif (TRM) (*Garber et al., 1998*; *Das et al., 2004*), increasing the order of the TRM in the quaternary complex. The TRM wraps around Tat, exposing several basic residues, and contributing CycT1 C261 to coordinate a shared $Zn^{2+}$ ion. Remarkably, AFF4 fragments containing the CycT1- and Tat-interacting segments increased the affinity of Tat-P-TEFb for TAR by 30-fold. These results support the idea that AFF4 contributes to TAR binding through two distinct and sequential mechanisms. AFF4 interactions with Tat directly favor SEC recruitment and AFF4 interactions that constrain the CycT1 TRM indirectly promote TAR binding.

## Results

The 2–73 fragment of the SEC scaffold protein, AFF4, binds with high affinity to P-TEFb and increases P-TEFb affinity for Tat 11-fold (*Chou et al., 2013*; *Schulze-Gahmen et al., 2013*). To define the structural basis for the increased affinity of the AFF4-P-TEFb complex for Tat, we determined the crystal

structure of the quaternary complex of P-TEFb with AFF4$_{2-73}$ and Tat$_{1-57}$. The Tat 1–57 fragment is a minimal construct with high transcriptional activation (*Garcia et al., 1988*). The structure was determined using X-ray data to 3.0-Å resolution (R/R$_{free}$ = 0.206/0.232; *Figure 1A*, *Table 1*) with three complexes in the asymmetric unit (a.u.).

Two complexes related by a non-crystallographic two-fold rotation axis are nearly identical, while the third complex shows small differences due to different crystal contacts (*Figure 1—figure supplement 1*). For example, only the two dyad-related complexes show electron density for AFF4 helix 0 (residues 4–21, *Figure 1A* and *Figure 1—figure supplement 3*), which packs against αE and αI of the CDK9 subunit in the same complex and AFF4 35–39 from the two-fold-related complex in the a.u. Almost identical interactions between AFF4 helix 0 and CDK9 were observed in one out of three assemblies in the AFF4-P-TEFb structure (*Schulze-Gahmen et al., 2013*), in addition to crystal contacts between helix 0 and a crystallographically related CDK9 subunit. This recurrence of the same

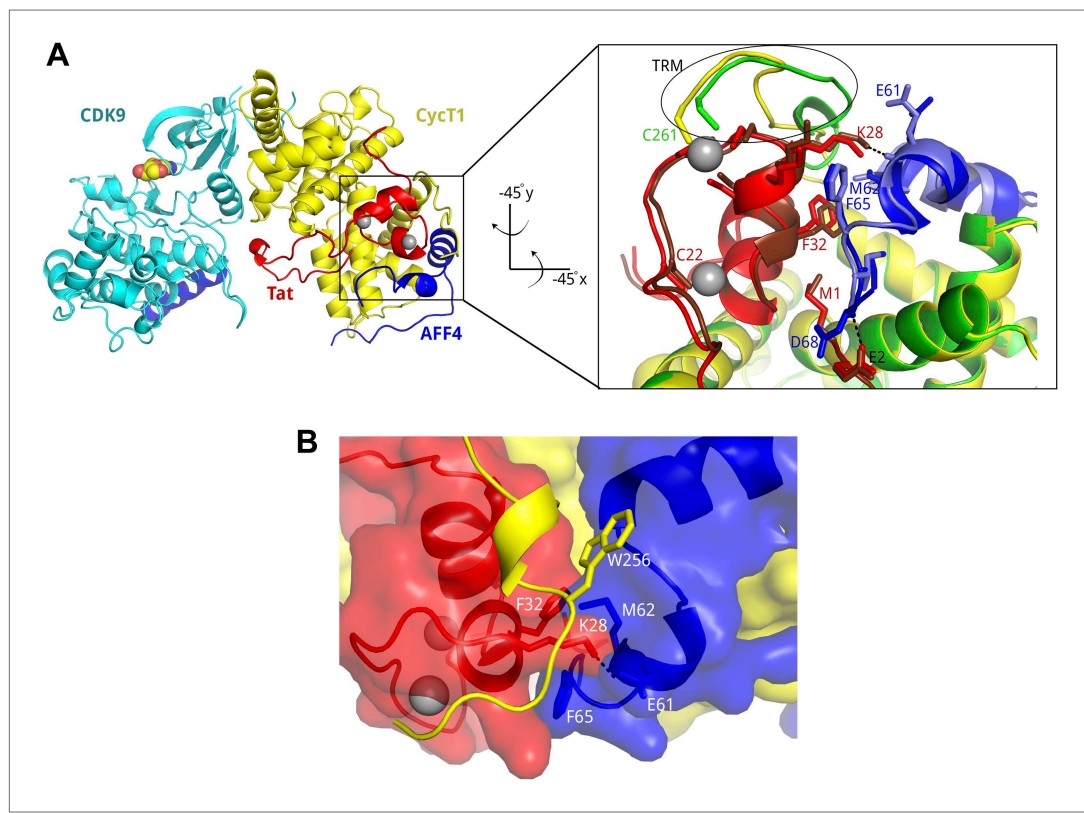

**Figure 1**. Tat and AFF4 bind adjacent to each other on the CycT1 surface. (**A**) Tat-AFF4-P-TEFb ribbon diagram (left) showing interactions between Tat (red) and AFF4 (blue) bound to the CycT1 (yellow) subunit of P-TEFb. AFF4 helix 0 is bound to the CDK9 (cyan) subunit, and adenosine (spheres) is modeled in the CDK9 ATP binding pocket. The close-up view (right) obtained by horizontal and vertical 45° rotations of the left hand figure shows similar Tat–AFF4 (red/dark red-blue/light blue) interactions in independent complexes. CycT1 (yellow/green) TRM residues adopt different structures in different crystal environments. The upper of the two Zn$^{2+}$ ions (gray spheres) anchors the CycT1 TRM. (**B**) Surface representation of the binding pocket for Tat K28 in the Tat–AFF4 (red-blue) interface. The CycT1 TRM (yellow ribbon) with the fewest crystal contacts is shown. The TRM interacts with a hybrid interface including AFF4 and Tat.

The following figure supplements are available for figure 1:

**Figure supplement 1**. Crystal contacts of CycT1 TRM for two representative complexes in the a.u..

**Figure supplement 2**. Surface representation of the binding pocket for Tat M1 and the N-acetyl group.

**Figure supplement 3**. Schematic drawing of AFF4 secondary structures.

**Table 1.** X-ray data collection and refinement statistics for P-TEFb-Tat-AFF4

| Data collection | |
|---|---|
| Space group | $P6_522$ |
| Cell dimensions: $a, b, c$ | 184.91, 184.91, 360.40 |
| Resolution (Å)* | 50.0–3.0 (3.05–3.0) |
| Unique reflections* | 73,424 (3589) |
| I/σ(I)* | 12.8 (0.9) |
| $R_{merge}$ (%)* | 22.2 (>100) |
| $R_{merge}$ (%)*, I/sigI≥3 | 8.4 (18.9) |
| $R_{pim}$ (%)† | 7.6 (87.9) |
| $CC_{1/2}$ high resolution shell | 0.553 |
| Completeness (%)* | 100.0 (100.0) |
| Redundancy* | 24.2 (23.8) |
| Temperature (K) | 100 |
| Mosaicity (°) | 0.23–0.39 |
| Refinement | |
| Resolution (Å) | 49.0–3.0 |
| No. reflections | 73,297 |
| $R_{work}/R_{free}$* | 0.206/0.232 (0.316/0.335) |
| No. atoms/B-factors (Å²) | |
| CDK9, molecule 1, 2, 3 | 2560 (75.4), 2521 (90.9), 2572 (88.5) |
| Cyclin T1, molecule 1, 2, 3 | 2061 (79.4), 2053 (85.8), 2058 (97.8) |
| AFF4$_{34-66}$, molecule 1, 2, 3 | 438 (85.0), 268 (115.7), 422 (92.3) |
| Tat | 390 (79.1), 384 (78.0), 390 (102.7) |
| Water | 37 (58.7) |
| R.m.s. deviations | |
| Bond lengths (Å) | 0.0035 |
| Bond angles (°) | 0.811 |
| Ramachandran plot‡ | |
| Favored (%) | 96.0 |
| Allowed (%) | 3.36 |
| Disallowed (%) | 0.66 |

*Values in parentheses are for the highest resolution shell.
†$R_{p.i.m.}$ is the precision-indicating merging R factor, which is related to the traditional $R_{sym}$ but provides a better estimate of data quality (**Weiss and Hilgenfeld, 1997**; **Weiss et al., 1998**).
‡Values from MOLPROBITY (**Chen et al., 2009**).

helical structure in different crystal environments suggests that AFF4 residues 4–21 prefer a helical conformation. The function of helix 0, however, remains in doubt because mutational effects on transcription do not match the AFF4 contacts in the interface, and stabilization of helix 0 depends on crystal packing (**Schulze-Gahmen et al., 2013**). We will focus on features shared among all complexes and point out differences when they are relevant for the discussion.

Tat binds in an extended conformation to AFF4-P-TEFb, with minor changes from the Tat-P-TEFb structure (**Tahirov et al., 2010**). The backbone of loop residues 27–30 shifts in response to contacts with CycT1 250–261 for two complexes in the a.u. Based on mass spectrometry and electron density, the baculovirus-expressed Tat, like Tat expressed in HEK293T cells (**Jäger et al., 2012**), is N-terminally acetylated (**Figure 1—figure supplement 2**). Acetylation of the Tat N-terminus removes a positive charge in the moderately hydrophobic Tat–CycT1 interface around Tat M1 and fills a cavity, probably leading to tighter anchoring of the Tat N-terminus to CycT1.

AFF4 residues 34- to 69-fold on the CycT1 surface, making multiple direct contacts with Tat K28, F32, and E2 (**Figure 1**) and burying Tat M1. The average size of the Tat–AFF4 interface is 305 Å² on Tat and 330 Å² on AFF4. The interactions between AFF4 and Tat are mostly hydrophobic and van der Waals contacts, but also include hydrogen bonds on each end of the interaction site (**Figure 1A**). The nexus of the Tat–AFF4 interface, Tat K28, is partially buried in a hydrophobic pocket formed by the side chains of AFF4 M62, F65, and Tat F32, and the main chain of AFF4 helix 2, residues 59–63. The Tat K28 side-chain amino group forms a hydrogen bond with the AFF4 E61 main-chain carbonyl at the pocket edge facing the solvent (**Figure 1A**). In addition, the Tat E2 side chain is positioned to form a hydrogen bond with the AFF4 D68 main-chain amide. The CycT1 TRM forms another side of the K28 pocket (**Figure 1B**).

Although AFF4 contains similar secondary structural elements observed in the absence of Tat (**Schulze-Gahmen et al., 2013**), coupled shifts occur to avoid collisions with Tat (**Figure 2**). Backbone RMS deviations in AFF4 residues 34–66 excluding the variable loop 43–45 range from 1.6 Å to 2.2 Å between AFF4-P-TEFb complexes with and without Tat. In contrast, RMS deviations for the same AFF4 residues of different complexes in the a. u. range from 0.32 Å to 0.58 Å. While AFF4 residues 34–40 coincide in the presence and absence of Tat, AFF4 helix 1 (residues 48–55) is shifted along the helix axis. This change positions AFF4 M55 to make contacts with L252 in the CycT1 TRM region (**Figure 2A**). AFF4 helix 2 (residues 58–66) is shifted away from the bound Tat and closer to the CycT1 surface formed by helices H2′, H3′ and the

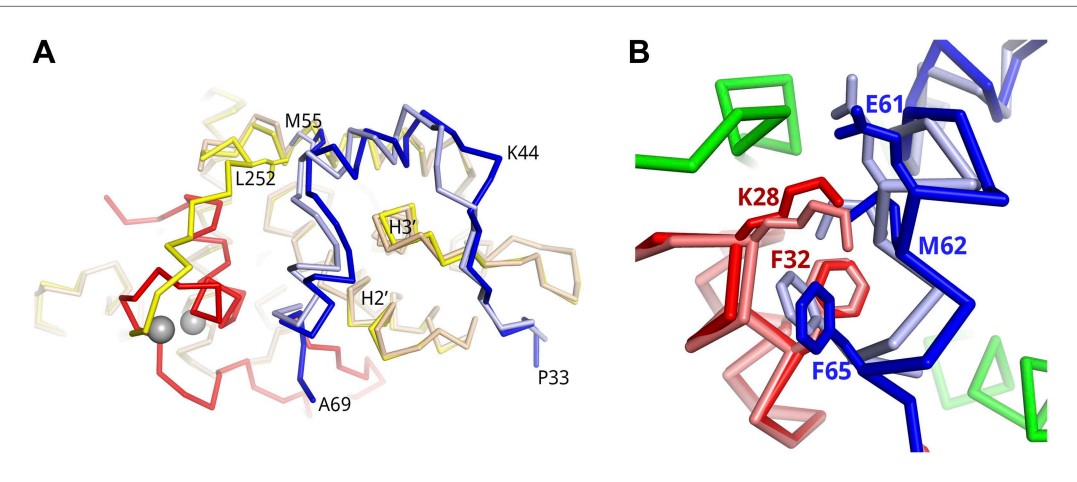

**Figure 2**. Structural shifts in the subunits of the Tat-AFF4-P-TEFb complex. (**A**) Superposition of the AFF4-P-TEFb complex (PDB ID 4IMY, pastel colors) and Tat-AFF4-P-TEFb (red, blue, yellow) on the CycT1 subunit shows coupled shifts of the two AFF4 helices. AFF4 helices 1 and 2 shift away from Tat, thereby avoiding close contacts between Tat and helix 2. (**B**) Superposition of AFF4-P-TEFb (PDB ID 4IMY, AFF4 light blue), Tat-P-TEFb (PDB ID 3MI9, Tat pastel-red), and Tat-AFF4-P-TEFb (Tat red, AFF4 blue, CycT1 green) on the CycT1 subunit. CycT1 of AFF4-P-TEFb and Tat-P-TEFb is omitted to emphasize the changes in AFF4. The AFF4 backbone shifts 1–2 Å in the presence of Tat, while the Tat conformation displays only small changes associated with AFF4 binding. Side chains undergo only small conformational changes.

H3′–H4′ loop. This movement leads to the formation of additional hydrogen bonds between AFF4 and CycT1 in the Tat-AFF4-P-TEFb complex compared to the AFF4-P-TEFb complex. As a consequence of the shifts in the AFF4 backbone, unfavorable close contacts between AFF4 and Tat are avoided (**Figure 2B**).

In contrast to the Tat-P-TEFb complex, in which the CycT1 TRM is disordered between residues 253 and 260, this functionally important segment is ordered in two conformations in the Tat-AFF4-P-TEFb complex (**Figures 1, 3**). In all three complexes in the a.u., P249 and N250 at the beginning of the TRM make multiple contacts with the main chain atoms at the C-terminal end of Tat helix 35–44. In addition, CycT1 L252 forms hydrophobic interactions with AFF4 M55 in all three complexes. The TRM structures start to diverge at this point. CycT1 residues 253–259 loop over Tat helix 28–33 in two conformations that converge at CycT1 C261 (**Figure 3A**, **Figure 3—figure supplement 1**). In the dyad-related complexes, the CycT1 TRM makes crystal contacts that include W258, R259, and A260, but in the third complex, the crystal contacts are restricted to the side chain of R251. In all complexes, W256 makes buried contacts with AFF4. Basic residues such as K253, R254, R259, and the polar N257, on the other hand, show continuous main-chain electron density but the exposed side chains are disordered. C261 binds the shared Tat $Zn^{2+}$ ion in all three complexes. The presence of multiple conformations and relatively weak electron density for the TRM loop residues 253–260 indicates that this region is conformationally restrained but still quite flexible after Tat and AFF4 binding. These results suggest that in the presence of Tat, AFF4 partially orders the TRM (**Figure 3B**).

The structure of the Tat-AFF4-P-TEFb complex points to AFF4 M62 and F65 as the major Tat-interacting residues. To test the contribution of the Tat–AFF4 interface to Tat-dependent transcription, we measured the effect of alanine substitutions on SEC recruitment and Tat-dependent HIV-1 transcription. These assays were performed with the AFF1 scaffold protein, because Tat has a stronger effect on HIV transcription with AFF1 than with AFF4 (**He et al., 2010**; **Lu et al., 2014**). AFF1 mutants V67A and F70A (corresponding to M62A and F65A in AFF4) were ectopically expressed in HeLa cells in the absence or presence of Tat(C22G), a mutation in the $Zn^{2+}$ ion coordination site required for WT Tat activity (**Garber et al., 1998**). The Tat(C22G) mutation increases the dependence on AFF1 for efficient transactivation (**Lu et al., 2014**). Immunoprecipitation of tagged WT AFF1, as well as the V67A, F70A, and V67A/F70A variants, efficiently co-precipitated CDK9 and CycT1 (**Figure 4A**). In contrast, Tat(C22G) failed to co-precipitate P-TEFb (**Figure 4B**, lanes 1 & 2). This lack of binding was rescued by co-expressing WT AFF1 but not the three AFF1 alanine variants, V67A, F70A, and V67A/

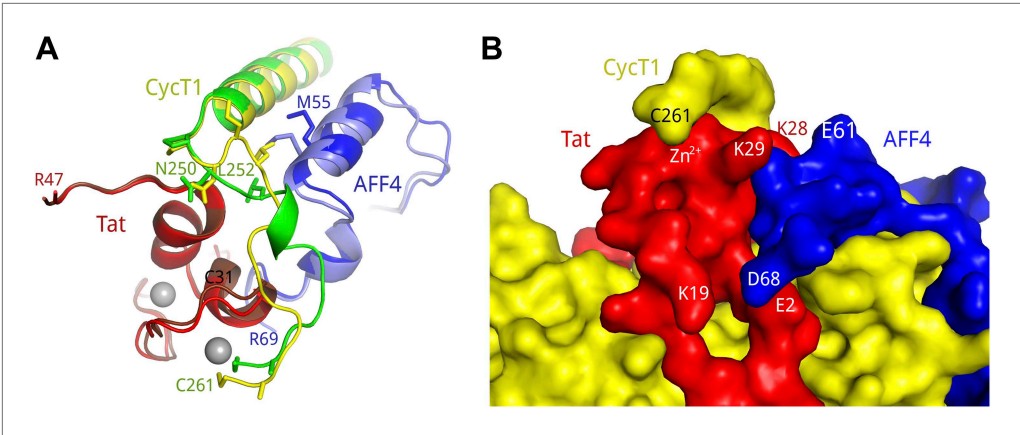

**Figure 3**. CycT1 TRM interacts with Tat and AFF4. (**A**) Ribbon diagram of two distinct TRM conformations observed in the Tat-AFF4-P-TEFb crystal structure (red, blue, yellow/dark red, light blue, green). $Zn^{2+}$ ions are shown as gray spheres. (**B**) Surface representation of Tat-AFF4-CycT1 interactions. The three subunits intertwine, thereby stabilizing the TRM conformation in the hybrid interface.

The following figure supplements are available for figure 3:

**Figure supplement 1**. Representative electron density for the Tat-AFF4-P-TEFb complex.

---

F70A (**Figure 4B**). These results suggest that AFF1 V67 and F70 are important for interactions between the scaffold and Tat. In turn, this interface stabilizes the AFF1-CycT1 association, as suggested by the Tat-AFF4-P-TEFb structure.

The effects of changes in the Tat–AFF1 interface on Tat-dependent HIV transcription were measured using a HIV LTR-driven luciferase reporter system in HeLa-derived NH1 cells (**He et al. 2010**) that can express Tat(C22G). Although Tat(C22G) barely activated HIV transcription by itself (1.9-fold), this mutant strongly synergized with WT AFF1 to stimulate transcription to a much higher level (89-fold). Ectopic expression of WT AFF1 by itself only increased Tat-independent luciferase expression 13-fold (**Figure 4C**). In contrast, the AFF1 single (V67A and F70A) and double (V67A/F70A) alanine mutants showed significantly reduced cooperation with Tat(C22G) in activating HIV transcription compared to WT AFF1 (**Figure 4C**). Thus, the Tat–AFF4 interface is critical not only to enhance the binding of Tat to CycT1, but also to stimulate Tat-dependent HIV transcription.

The Tat–TAR recognition motif of CycT1 is essential for high affinity binding of P-TEFb-Tat to TAR (**Garber et al., 1998**). This critical segment (CycT1 250-264) at the C-terminal end of the cyclin domain interacts directly with TAR, as judged by RNA–protein cross-linking studies (**Richter et al., 2002**). Since AFF4 binds close to the TRM in the Tat-AFF4-P-TEFb structure, we investigated the effect of AFF4 on TAR binding. Electrophoretic mobility shift assays (EMSA) revealed unexpectedly that AFF4 fragments 32–67, 2–73, and 2–98 each increased the affinity of TAR for the Tat-P-TEFb complex by 30-fold (**Figure 5A**). It is unlikely that AFF4 directly contributes to TAR binding, since the AFF4-P-TEFb complex does not show any binding to TAR by itself. Instead, the Tat-AFF4-P-TEFb structure provides evidence that AFF4 binding in the presence of Tat restricts the conformational freedom of the CycT1 TRM region and positions this region for TAR interaction (**Figure 5B**).

## Discussion

Efficient HIV transcription requires the Tat/TAR-mediated recruitment of SECs to the HIV promoter. Transcription remains stalled at the promoter in the absence of host elongation factors needed for Pol II to faithfully reach the distal end of the HIV genome. The X-ray structure of Tat-AFF4-P-TEFb reveals subunit interactions that mediate the preference of Tat for SECs over other P-TEFb complexes. As predicted (**Schulze-Gahmen et al., 2013**), Tat and AFF4 bind adjacent to each other on the CycT1 surface (**Figure 1**). The Tat–AFF4 interaction surface is centered around Tat K28, which is acetylated in vivo to regulate HIV transcription (**Kiernan et al., 1999**). Mutations in AFF1 corresponding to the Tat–AFF4 interface in the crystal structure reduce scaffold binding and transcription stimulation functions.

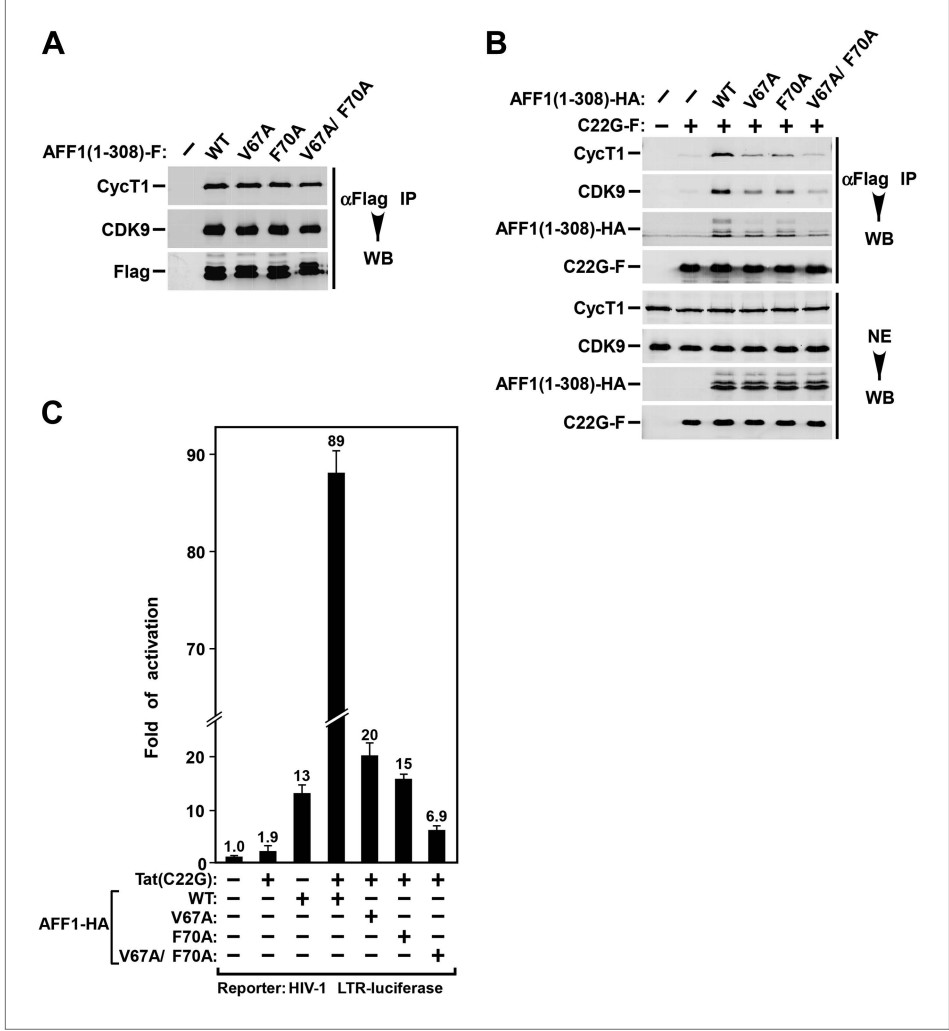

**Figure 4**. AFF1 Tat interaction mutants reduce Tat binding and activation of HIV LTR by AFF1. (**A**) Nuclear extracts (NE) were prepared from HeLa cells expressing the truncated Flag-tagged AFF1 protein (residues 1–308). Anti-Flag immunoprecipitates (IP) from the NE were examined by Western blotting (WB) for the indicated proteins. (**B**) Nuclear extracts were prepared from HeLa cells co-expressing Flag-tagged Tat(C22G) and haemagglutinin (HA)-tagged truncated AFF1. Anti-Flag IPs were analyzed as in **A**. (**C**) HeLa-based NH1 cells containing the intergrated HIV-1 LTR-luciferase reporter gene were transfected with the Tat(C22G)- and/or AFF1-expressing construct as labeled. Luciferase activities were measured in cell extracts, with the level of activity detected in cells transfected with an empty vector (−) set to 1. The error bars represent mean ± SD from three independent measurements.

Compared to the AFF4-P-TEFb complex, AFF4 structural segments in the Tat-AFF4-P-TEFb complex undergo unanticipated 1–2 Å backbone shifts to avoid unfavorable close contacts with Tat (*Figure 2*). These shifts in AFF4 alter the dimensions of the Tat-binding pocket in AFF4-P-TEFb that may serve as a site for targeting HIV transcription inhibitors (*Schulze-Gahmen et al., 2013*).

The Tat-AFF4-P-TEFb complex structure also reveals that the scaffold and Tat combine to partially fold the CycT1 TRM. This interaction is an example of multiple natively disordered protein segments coming together to form a structure that depends on the other protein subunits (*Figure 3B*). The conformations of the CycT1 TRM are strikingly different in the Tat-AFF4-P-TEFb complex compared to other structures containing P-TEFb. The TRM residues 253–260 are disordered in the Tat-P-TEFb crystal structure (*Tahirov et al., 2010*). In P-TEFb alone, crystal contacts stabilize the TRM in a distinct conformation that partially occludes the AFF4 binding site and precludes contacts between C261 and the Tat $Zn^{2+}$ ion (*Garber et al., 1998*; *Baumli et al., 2008*).

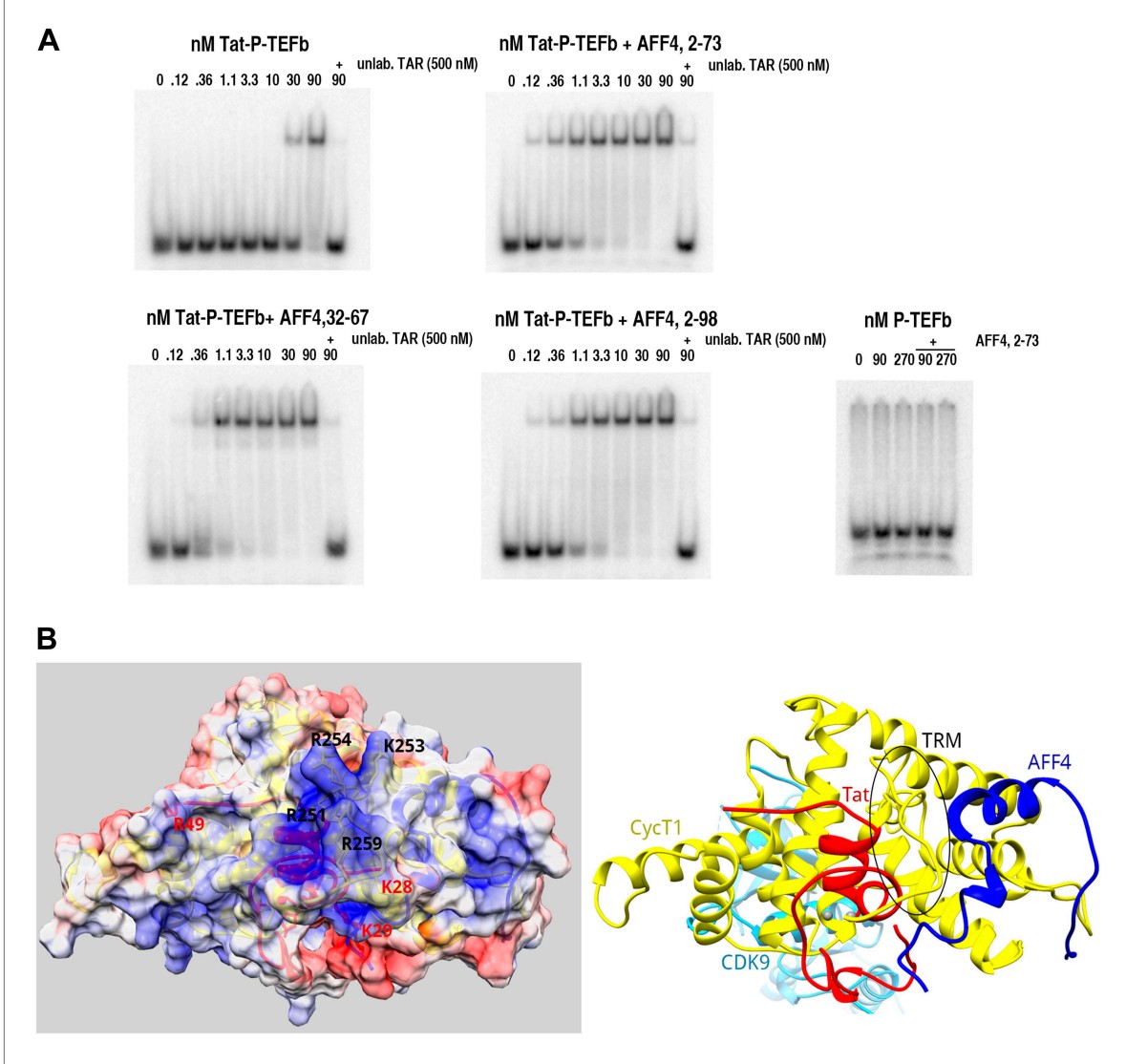

**Figure 5**. SECs stimulate TAR recognition. (**A**) Electrophoretic mobility shift assays with $^{32}$P-labeled TAR and increasing concentrations of Tat-P-TEFb, or Tat-P-TEFb + AFF4$_{32–67}$, Tat-P-TEFb + AFF4$_{2–73}$, Tat-P-TEFb + AFF4$_{2–98}$. Control assays (bottom right) with P-TEFb and AFF4$_{2–73}$ showed no shifts for TAR. Half of TAR was shifted with 35–40 nM Tat-P-TEFb complex. In the presence of excess AFF4 fragments 32–67, 2–73, or 2–98, 50% of TAR was shifted by 1.1 nM Tat-AFF4-P-TEFb complex. (**B**) Calculated electrostatic surface potential of Tat-AFF4-CycT1 centered on the CycT1 TRM. The ribbon diagram (right) is in the same orientation as the surface representation (left). This orientation converts into the orientation in *Figure 1A* by consecutive rotations around y (70°) and z (−35°). CDK9 was omitted from the surface figure (left) to focus on the TAR interaction region. Solvent-exposed CycT1 residues K253, R254, N257, W258, R259, and Tat R49, which have no side-chain electron density, were modeled in the most common orientation. The electrostatic potential, calculated using APBS (*Baker et al., 2001*) was applied to color the solvent excluded surface of Tat-AFF4-CycT1 in Chimera (*Pettersen et al., 2004*) from −5 k$_b$Te$^{−1}$ (red) to +5 k$_b$Te$^{−1}$ (blue). CycT1 residues were labeled in black, Tat residues in red. The TRM region forms a positively charged patch on the SEC surface close to the disordered Tat ARM, which follows Tat R49.

The following figure supplements are available for figure 5:

**Figure supplement 1**. Model of TAR binding to SEC.

The position of the TRM in the Tat-AFF4-P-TEFb structure, including exposed side chains for residues R251, R254, W258, and R259, is consistent with functional data (*Garber et al., 1998*). For example, alanine mutants of eight out of thirteen residues in the CycT1 segment 250–262 abolished or reduced in vitro binding of Tat-CycT1 to TAR. In addition, UV crosslinking of the CycT1 Tat–TAR complex (in the absence of AFF4) suggests that TAR makes direct contacts with the TRM (*Richter et al., 2002*).

The CycT1 TRM is not known to make RNA contacts during host transcription in uninfected cells, but the TRM contacts the loop in HIV TAR (*Wei et al., 1998*; *Richter et al., 2002*). To probe the complementarity of TAR with the Tat–CycT1 hybrid surface in the Tat-AFF4-P-TEFb complex structure, we manually docked the TAR-argininamide solution structure (PDB ID 1ARJ) (*Puglisi et al., 1992*; *Aboulela et al., 1995*) with Tat-AFF4-P-TEFb. Since, we could not model the side chains of CycT1 K253, R254, W258, and R259 in the electron density, we added these side chains in the most frequently observed conformation. The calculated electrostatic potential surface shows a large positively charged region covering the CycT1 TRM and part of Tat including R49 (*Figure 5B*). Positioning the TAR loop to contact the TRM enables the RNA bulge (U23–U25) to reach the Tat arginine rich motif (ARM) that makes essential contacts (*Figure 5—figure supplement 1*) (*Weeks and Crothers, 1991*). The dimensions of all components fit well and provide a plausible working model for the TAR interaction with Tat-AFF4-P-TEFb.

AFF4 not only promotes folding of the TRM, but the scaffold also increases TAR binding in vitro by 30-fold. This enhancement of RNA affinity is likely to be an indirect consequence of ordering the CycT1 TRM, because TAR is too small to make direct contacts with AFF4 residues in the complex. In Hela cells, alanine mutations in AFF1 (M60A/L61A) corresponding to the AFF4–TRM interaction site (M55/L56) abolish Tat(C22G)-SEC assembly and eliminate the inhibitory activity of AFF1 1-308 in Tat transactivation (*Lu et al., 2014*). Taken together, both the Tat-AFF4 interface and the TRM–AFF4 interaction are essential for WT Tat activity.

These results show that AFF4 contributes to the selective recruitment of SECs by Tat and TAR through a two-stage mechanism. First, direct interactions involving AFF4, Tat, and the TRM increase the binding affinity of Tat for AFF4-P-TEFb 11–fold (*Schulze-Gahmen et al., 2013*). Second, AFF4 binding to Tat-P-TEFb indirectly constrains the TRM conformation, increasing the binding affinity for TAR 30-fold. The enhancements of affinity in these successive steps together lead to a 330-fold increase in Tat/TAR binding by AFF4-P-TEFb over P-TEFb. This preference of Tat and TAR for the SECs, in concert with the Tat-stimulated release of scaffold-P-TEFb complexes from the 7SK snRNP (*Lu et al., 2014*), ensures preferential, simultaneous recruitment of the full complement of elongation factors required for efficient HIV transcription.

## Materials and methods

### Protein expression
P-TEFb and TAT-P-TEFb were expressed in High5 insect cells using recombinant baculovirus infections. We co-expressed human CDK9 1–330 and human cyclin T1 1–264 with and without HIV-1 Tat 1–57. Baculovirus generation and High5 cell infections were described in detail previously (*Schulze-Gahmen et al., 2013*). AFF4 fragments 2–73 and 2–98 with an N-terminal TEV-protease-cleavable His-tag were expressed in *E. coli* (*Schulze-Gahmen et al., 2013*).

### Purification of the Tat-AFF4-P-TEFb complex
Tat-P-TEFb and AFF4$_{2-73}$ were purified separately following procedures described recently (*Schulze-Gahmen et al., 2013*). Tat-P-TEFb and AFF4$_{2-73}$ were combined at a 1:1.4 (mol/mol) ratio, concentrated to 0.6 ml, and injected onto an analytical Superdex S200 size exclusion column equilibrated with 25 mM Na-HEPES pH 7.4, 0.2 M NaCl and 1 mM DTT. The center fractions of the eluted four-protein peak were used for crystallization.

### TAR RNA
A synthetic TAR fragment encompassing nucleotides 18–44 was purchased from IDT (San Diego, CA, USA). The RNA was annealed at 0.1 mg/ml in 20 mM Na HEPES pH 7.3, 100 mM KCl, 3 mM MgCl$_2$. Best results were obtained by heating the RNA at 75°C for 2 min, followed by rapid cooling on ice. The purity of the RNA, analyzed by denaturing and native 10% polyacrylamide gel electrophoresis, was at least 95%.

### Crystallization and structure determination
The purified Tat-AFF4-P-TEFb complex was combined with refolded synthetic TAR in a 1.1-fold molar excess. MgCl$_2$ was added to a 3 mM final concentration. The protein–RNA complex was concentrated in an Amicon Ultra filter with a 30 kDa cutoff to about 10 mg/ml protein concentration. The presence of TAR was confirmed on silver-stained polyacrylamide gels.

Crystals were grown in sitting drops from 1.0 µl protein–TAR complex combined with 1.0 µl reservoir solution. The drops were equilibrated against 2.4 M sodium formate, 10 mM MgCl$_2$ at 18°C. After equilibrating for 24 hr, diluted microseeds from previous crystallization experiments were added with a hair. Seeding produced single hexagonal crystals (0.15 × 0.15 × 0.2 mm).

Crystals were soaked in 2.8 M sodium formate, 10 mM MgCl$_2$, 30% glycerol for cryoprotection and flash frozen in liquid nitrogen. X-ray data were collected at Beamline 8.3.1 at the Advanced Light Source at the Lawrence Berkeley National Laboratory (**MacDowell et al., 2004**). The reflections were processed using HKL2000 (**Otwinowski and Minor, 1997**) (**Table 1**). The R$_{merge}$ of the data is high, requiring additional tests of the symmetry of the crystals. Processing the data in space group P1 yielded an R$_{merge}$ of 18.6%, and analyzing these data using the program Pointless (**Winn et al., 2011**) confirmed the presence of the symmetry operators of space groups P6$_5$22 or P6$_1$22. Moreover, no twinning was detected using the programs Pointless and Xtriage (**Adams et al., 2010**). R$_{merge}$ for the P6$_5$22 data set decreases to 8.4% after excluding reflection with I/sigI <3. In addition, the values for R$_{p.i.m.}$ (**Weiss and Hilgenfeld, 1997**; **Weiss et al., 1998**) and CC$_{1/2}$ (**Karplus and Diederichs, 2012**), as well as the refinement statistics indicate that the data were processed correctly. These tests support the conclusion that the space group was assigned correctly, and the unusually high value of R$_{merge}$ was due to the presence of many weak reflections and the high redundancy of the data set.

The structure was determined by molecular replacement with PHENIX (**Adams et al., 2010**) using the AFF4-P-TEFb complex (PDB ID 4IMY) as the search model. After three molecules were placed in the a. u., the Tat structure from the P-TEFb-Tat complex (PDB ID 3MI9) was combined with the molecular replacement solution by superimposing the CycT1 molecules of the different complexes. The model was refined with PHENIX.refine (**Adams et al., 2010**), using gradient minimization with weight optimization, maximum likelihood targets, non-crystallographic symmetry constraints, individual B-factors, and TLS parameters. Automatic refinement was alternated with manual rebuilding using Coot (**Emsley and Cowtan, 2004**).

Although the crystallization experiments were set up with Tat-AFF4-P-TEFb-TAR complex, we did not find electron density for TAR RNA, nor was there room for TAR in the crystal lattice. The high-salt conditions of the crystallization probably dissociated the TAR RNA from the protein complex. In the final model, density was missing for residues 1–7 and 89–96 in CDK9 mol1 and mol2, and residues 1–7 and 92–95 in CDK9 mol3. Density was also absent for residues 1–6 and 262–264 in all three CycT1 molecules, and residues 50–57 in all three Tat molecules. For AFF4 mol1 (mol3) density for residues 2, 22–32, 70–73 (2–3, 22–33, 70–73) was missing, while AFF4 mol2 was missing density for residues 2–33 and 70–73. The ATP binding pocket of CDK9 contained extra density although ATP was not included in the crystallization. The density was modeled as adenosine.

## Structure analysis

Least squares fitting of protein structures were performed with Coot (**Emsley and Cowtan, 2004**) and the program ProFit by Dr A Martin from University College London. Profit uses the McLachlan fitting algorithm (**McLachlan, 1982**). Potential hydrogen bonds were identified with the program CONTACT in CCP4 (**Winn et al., 2011**) and manually inspected.

## Co-immunoprecipitation assay

The assay was performed as described (**He et al., 2010**). Briefly, nuclear extracts prepared from HeLa cells transfected with the indicated expression constructs were incubated with anti-Flag or anti-HA agarose beads (Sigma, St. Louis, MO) for 2 hr before extensive washing and elution.

## Luciferase assay

The HeLa-based NH1 cell line containing an integrated HIV-1 LTR-luciferase reporter construct (**He et al., 2010**) was transfected with the indicated expression constructs. At 48 hr post transfection, total cell lysates were prepared from approximately 10$^6$ cells per sample and luciferase activity was measured.

## Electrophoretic mobility shift assay

Refolded synthetic TAR (nucleotides 18–44) was radioactively labeled with $^{32}$P-γ–ATP using T4-polynucleotide kinase. A 10-µl reaction was prepared with 200 nM TAR, 0.3 mCi $^{32}$P-γ–ATP (7000 Ci/mmol, MP Biomedicals, Sohon, OH), and 10 units of T4-polynucleotide kinase (New England BioLabs, Ipswich, MA) in 70 mM Tris/HCl pH7.6, 10 mM MgCl$_2$, 2 mM DTT. After incubating at 37°C for 1 hr, 25 µl of annealing buffer (20 mM Na HEPES pH 7.3, 100 mM KCl, 3 mM MgCl$_2$) were added

to the reaction. The mixture was purified twice over Illustra G25 spin columns (GE Healthcare, Piscataway, NJ) to remove free nucleotides. The purified labeled TAR was diluted to 10 nM (3000–5000 cpm/ μl) with annealing buffer for storage and use in EMSAs.

Binding reactions (10 μl) were carried out in 20 mM Na HEPES pH 7.3, 100 mM KCl, 3 mM $MgCl_2$, 1 mM DTT, 4% glycerol with 12 units RNasin (Promega, Madison, WI), 10 μg/ml BSA, and 5 μg/ml poly(I:C) (Invivogen, San Diego, CA). Each reaction contained 100 pM labeled TAR RNA. Reactions were incubated at 20°C for 30 min and RNA-binding complexes were separated on a pre-run 6% polyacrylamide gel in 0.5x TBE (100 V, 1 hr at 4°C). Gels were dried, exposed to storage phosphor screens, and measured on a Typhoon phosphorimager (GE Healthcare, Piscataway, NJ).

## Acknowledgements

Ann Fisher helped with baculovirus production, Yun Bai, and George Katibah provided valuable advice for RNA handling and EMSAs. James Holton aided in the confirmation of the space group symmetry. We are grateful to James Holton, George Meigs, and Jane Tanamachi at beamline 8.3.1 at Lawrence Berkeley National Laboratory for help with X-ray data collection. We appreciate the encouragement of the HARC Center.

## Additional information

### Funding

| Funder | Grant reference number | Author |
| --- | --- | --- |
| National Institutes of Health | P50GM82250 | Tom Alber |
| National Institutes of Health | R01AI095057, R01AI041757 | Qiang Zhou |
| California HIV/AIDS Research Program | ID09-B-026 | Tom Alber |

The funders had no role in study design, data collection and interpretation, or the decision to submit the work for publication.

### Author contributions

US-G, HL, Conception and design, Acquisition of data, Analysis and interpretation of data, Drafting or revising the article; QZ, TA, Conception and design, Analysis and interpretation of data, Drafting or revising the article

## Additional files

### Major dataset

The following dataset was generated:

| Author(s) | Year | Dataset title | Dataset ID and/or URL | Database, license, and accessibility information |
| --- | --- | --- | --- | --- |
| Alber T, Schulze-Gahmen U | 2014 | AFF4 binding to P-TEF-Tat indirectly stimulates TAR recognition of super elongation complexes at the HIV promoter | 4OGR; http://www.rcsb.org/pdb/search/structidSearch.do?structureId=4OGR | Publicly available at the RCSB Protein Data Bank (http://www.rcsb.org/pdb/). |

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
