## [Decision Letter]

Thank you for sending your work entitled ”AFF4 binding to Tat-P-TEFb indirectly stimulates TAR recognition of super elongation complexes at the HIV promoter” for consideration at *eLife*. Your article has been favorably evaluated by a Senior editor and 3 reviewers, one of whom, Wes Sundquist, is a member of our Board of Reviewing Editors.

The Reviewing editor and the other reviewers discussed their comments before we reached this decision, and the Reviewing editor has assembled the following comments to help you prepare a revised submission.

Summary and overall evaluation:

The authors report the crystal structure of a complex formed between HIV Tat (residues 1-57), CycT1:CDK9, and an interacting fragment from the super elongation scaffold, AFF4 (residues 2-73). The structure reveals the molecular basis by which AFF4 contributes to Tat-dependent elongation of viral transcription. In broad terms, the new structure was anticipated by the 2013 *eLife* paper from the same group, but the data are generally of high quality and there are some important new insights that merit visibility. In particular, the new work indicates that AFF4 stabilizes HIV elongation complex formation by two coupled mechanisms: (i) AFF4 contacts Tat on the Cyclin T1 surface to stabilize protein complex formation. (ii) AFF4 indirectly promotes binding of the TAR RNA by stabilizing a Cyclin T1 TRM conformation that is competent for TAR binding. Important aspects of the structure are confirmed by mutagenesis and binding or Tat transactivation assays, and the positive influence of AFF4 on the affinity of Tat:P-TEFb complex for TAR RNA is confirmed by EMSA. The paper makes important contributions to the field and is appropriate for publication in *eLife*.

Significant issues that should be addressed:

The Rsym values are high, but as noted by the authors this widely used metric is flawed whereas the more reasonable Rpim and CC1/2 values support the crystallographic analysis. The authors argue that the high redundancy in the data set should allow them to recover meaningful information in the highest resolution shell, but they should report the final shell Rwork/Rfree to allow readers to judge how well the model matches the data at the highest resolution.

Issues for the authors' consideration:

1) The structure is with AFF4 but the functional data were obtained with AFF1, which the authors note is much more active than AFF4. The original AFF protein cloned by the Roeder lab as a CycT1-binding protein was a strong inhibitor of Tat transactivation, so different family members clearly have very different abilities to stimulate Tat. Can the authors explain the differential activity of AFF4 vs AFF1? Does the difference result from differential AFF4/AFF1 associations with other SEC proteins, or is AFF4 binding to CycT1 or Tat altered? If the latter is true, then important features could be missing in the AFF4 structure and should at least be commented upon.

2) It's often hard to get oriented in the different figures. To improve clarity, the authors should considering choosing some standard view(s) or at least relate the views they choose to the overview Figure 1.

---

## [Author Response]

*Significant issues that should*
*be addressed:*

*The Rsym values are high, but as noted by the authors this widely used metric is flawed whereas the more reasonable Rpim and CC1/2 values support the crystallographic analysis. The authors argue that the high redundancy in the data set should allow them to recover meaningful information in the highest resolution shell, but they should report the final shell Rwork/Rfree to allow readers to judge how well the model matches the data at the highest resolution*.

We shared the concern that the Rsym value is unusually high, possibly reflecting twinning or an incorrect assignment of the crystal symmetry. To confirm the space group, we processed the data in lower symmetry point groups, including P3 and P1. The Rsym values were not significantly lowered in any of these data sets. Analysis with the programs Pointless and Xtriage confirmed the presence of the expected symmetry operators and detected no characteristics of twinning. The Rsym for the P6522 data with I/sigI >3 was 8.4%, which is a more acceptable value. A discussion of these calculations, which strengthen the manuscript by supporting the space group assignment, was added to the Materials and methods. We also added values of Rwork/Rfree for the highest resolution shell.

*Issues for*
*the authors' consideration:*

*1) The structure is with AFF4 but the functional data were obtained with AFF1, which the authors note is much more active than AFF4. The original AFF protein cloned by the Roeder lab as a CycT1-binding protein was a strong inhibitor of Tat transactivation, so different family members clearly have very different abilities to stimulate Tat. Can the authors explain the differential activity of AFF4 vs AFF1? Does the difference result from differential AFF4/AFF1 associations with other SEC proteins, or is AFF4 binding to CycT1 or Tat altered? If the latter is true, then important features could be missing in the AFF4 structure and should at least be commented upon*.

The question of promoter specificity of AF-4 family members is a profound issue that is the subject of active investigation in multiple laboratories. Several hypotheses are being tested in these studies. These models include the idea that SEC components other than P-TEFb influence the activity of complexes containing different AFF scaffolds. In addition, evidence suggests that different scaffolds are recruited with different efficiency to different gene promoters (e.g., Lin C et al. (2011) *Genes Dev*
**25**: 1486; Liu P et al. (2014) *J Biol Chem* jbc.M113.539015). Posttranslational modifications or distinct allosteric transitions in partner proteins may also distinguish different scaffolds. These and other models must be tested to explain the higher activity of AFF1 compared to AFF4 at the HIV promoter. Due to the broad scope of this question and the limited insights into promoter selectivity provided by our data, we prefer not to speculate about this question in this manuscript.

*2) It's often hard to get oriented in the different figures. To improve clarity, the authors should considering choosing some standard view(s) or at least relate the views they choose to the overview*
Figure 1.

Two key figures were improved to address this point. Figure 1 were combined and the relative orientation of the views was indicated. In addition, a ribbon diagram was added to Figure 5 to orient the reader and better indicate the position of the TAR binding site.